# Development of a Transformation Method for Metschnikowia borealis and other CUG-Serine Yeasts

**DOI:** 10.3390/genes10020078

**Published:** 2019-01-23

**Authors:** Zachary B. Gordon, Maximillian P.M. Soltysiak, Christopher Leichthammer, Jasmine A. Therrien, Rebecca S. Meaney, Carolyn Lauzon, Matthew Adams, Dong Kyung Lee, Preetam Janakirama, Marc-André Lachance, Bogumil J. Karas

**Affiliations:** 1Designer Microbes Inc., London, ON N6G 4X8, Canada; zgordon2@uwo.ca (Z.B.G.); cleichth@uwo.ca (C.L.); jasmine.alyssa.therrien@gmail.com (J.A.T.); rmeaney2@uwo.ca (R.S.M.); carolyn.lauzon@gmail.com (C.L.); adams.mil@hotmail.com (M.A.); preetam.janakirama@gmail.com (P.J.); 2Department of Biochemistry, Schulich School of Medicine and Dentistry, University of Western Ontario, London, ON N6A 5C1, Canada; 3Department of Biology, University of Western Ontario, London, ON N6A 5B7, Canada; msoltys4@uwo.ca (M.P.M.S); dlee335@uwo.ca (D.K.L.); lachance@uwo.ca (M.-A.L.)

**Keywords:** genome engineering, synthetic biology, yeasts, *Metschnikowia*, genetic tools, DNA delivery, CUG-Ser

## Abstract

Yeasts belonging to the *Metschnikowia* genus are particularly interesting for the unusual formation of only two needle-shaped ascospores during their mating cycle. Presently, the meiotic process that can lead to only two spores from a diploid zygote is poorly understood. The expression of fluorescent nuclear proteins should allow the meiotic process to be visualized in vivo; however, no large-spored species of *Metschnikowia* has ever been transformed. Accordingly, we aimed to develop a transformation method for *Metschnikowia borealis*, a particularly large-spored species of *Metschnikowia*, with the goal of enabling the genetic manipulations required to study biological processes in detail. Genetic analyses confirmed that *M. borealis*, and many other *Metschnikowia* species, are CUG-Ser yeasts. Codon-optimized selectable markers lacking CUG codons were used to successfully transform *M. borealis* by electroporation and lithium acetate, and transformants appeared to be the result of random integration. Mating experiments confirmed that transformed-strains were capable of generating large asci and undergoing recombination. Finally, random integration was used to transform an additional 21 yeast strains, and all attempts successfully generated transformants. The results provide a simple method to transform many yeasts from an array of different clades and can be used to study or develop many species for various applications.

## 1. Introduction

For decades, yeasts have proven to be tremendously useful in the study of life’s fundamental processes. As single-celled models for eukaryotic life, they have short generation times, can be easily manipulated, and contain thousands of conserved genes—many of which are vitally important across nearly all living organisms [1,2]. While the discovery of new species provides us with new opportunities, genetic tools previously developed for *Saccharomyces cerevisiae* and other model organisms must be re-optimized for use in newly discovered species [3,4]. Altered membrane composition, modified gene expression, and alternative codon usage can prevent the seamless application of established transformation methods onto new species. Such transformation methods are crucial in the development of targeted knockouts, changes in gene expression, and fusion proteins that allow the study of biological processes in detail.

Of particular interest among yeasts are the 81 or more species assigned to the genus *Metschnikowia*, some of which have become popular for their varied applications [5]. For example, *M. pulcherrima* is of importance to winemakers for its generation of pleasant aromas in high-quality wines, and its anti-microbial properties further its potential as an industrial microorganism [6,7]. Similarly, *M. bicuspidata* has been studied extensively for its role as a pathogen, and other species (including *M. agaves* and *M. hawaiiensis*) are being studied as potential anti-aging agents in cosmetics [5].

*Metschnikowia* species are particularly intriguing, due to the unusual formation of only two needle-shaped ascospores during their sexual cycle. A two-spored meiotic product stands in sharp contrast to the four spores arising from meiosis in many other yeast species (like *S. cerevisiae*) [5]. Although the occurrence of meiotic recombination has been demonstrated in some *Metschnikowia* species [8] the fate of the nuclei during meiosis remains unclear, and is yet to be elucidated. The expression of fluorescent nuclear fusion proteins should allow the meiotic process to be visualized in vivo, but the generation of such fusions is greatly hindered by the lack of genetic tools available for *Metschnikowia* species. *M. pulcherrima* is the only species ever to have been transformed [9]. Unfortunately, the small size of its ascospores and the difficulty of generating abundant asci make *M. pulcherrima* less than ideal for the study of meiotic nuclei.

In contrast, *M. borealis* is one of many *Metschnikowia* species that form ascospores that can reach 20–50 times the size of normal budding cells [10]. The giant, elongated spores would greatly facilitate the task of elucidating the meiotic process in *Metschnikowia* species. The initial selection of *M. borealis* for study is further justified by the fact that the species has the highest maximum growth temperature among the large-spored *Metschnikowia* species (37 °C), making it more likely to survive the high-temperature incubations required by many of the conventional yeast transformation methods [11,12]. Therefore, we aimed to develop and optimize a transformation method for *M. borealis* to enable the study of the species in more detail.

Our approach builds on the conventional methods used to transform other yeast species, such as *S. cerevisiae* and *M. pulcherrima*. After identifying antibiotics that can be used as selectable markers, we developed various genetic constructs carrying the appropriate genes to allow growth on selective plates. We then tested conventional transformation techniques, such as lithium acetate transformation and electroporation to introduce the various genetic cassettes [11]. It was originally anticipated that differences in membrane composition, antibiotic sensitivity, and origins of replication may pose significant obstacles in our ability to seamlessly apply classical transformation methods from *S. cerevisiae* onto *M. borealis*. Accordingly, we developed a systematic approach entailing a variety of different transformation methods, using a variety of different antibiotic resistance markers to develop a successful transformation method for *M. borealis* and other CUG-Ser yeasts.

## 2. Materials and Methods

### 2.1. Microbial Strains and Growth Conditions

*Metschnikowia borealis* strains UWOPS 96-101.1 (*MATα*) and SUB 99-207.1 (*MATa*), and other strains that were used in transformation experiments were obtained from the yeast collection of the Department of Biology, University of Western Ontario, where they are kept frozen in liquid nitrogen. *Saccharomyces cerevisiae* VL6-48 (ATCC MYA-3666: *MATα his3-Δ200 trp1-Δ1 ura3-52 lys2 ade2-1 met14 cir^0^*). All strains were grown in YPAD broth (1% yeast extract, 2% peptone, 0.01% adenine hemisulfate, 2% d-glucose) at 30 °C with shaking at 225 rpm, with the exception of *C.* aff. *bentonensis*, M. *bicuspidata* and *M. orientalis*, which were grown at 27 °C. Stationary cultures were grown on the same media containing 1% agar at the same growth temperatures.

### 2.2. Codon-Optimization of KanMX and Sh ble

To codon-optimize the KanMX and Sh *ble* selectable markers, each CTG codon was changed to a TTG, and the resulting gene sequence was uploaded to the OPTIMIZER tool (http://genomes.urv.es/OPTIMIZER/) [13]. The codon frequencies from *M. borealis* in three highly expressed yeast genes—*ADH1*, *TDH3*, and *ENO1*—were used to guide optimization. A table was generated using the Codon Usage Database (https://www.kazusa.or.jp/codon/) and entered into OPTIMIZER to guide codon-optimization. The method used for optimization was one amino acid to one codon, with a maximum of 25 allowed codon changes. Optimized genes were called *MbKanMX* and *MbShBle*.

### 2.3. Yeast DNA Isolation

Genomic DNA was isolated from *M. borealis* using a modified alkaline lysis method. Individual colonies were suspended in 1.0 mL of sterile double-deionized water (sddH_2_O) and pelleted at 3000× *g* for 5 min. Cells were resuspended in 250 μL P1 Buffer (Qiagen, Valencia, CA, USA), 0.25 μL of 14 M β-mercaptoethanol, and 12.5 μL zymolyase solution (10 mg mL^−1^ zymolyase 20T in 25% glycerol), and incubated at 37 °C for 1 h. Following incubation, 250 μL of P2 Buffer (Qiagen) was added, gently mixed, and the suspension was incubated at room temperature for 5 min. Subsequently, 250 μL P3 Buffer (Qiagen) was added, gently mixed, and the suspension was incubated on ice for 10 min. Cellular debris was pelleted at max speed for 10 min at 4 °C, and the supernatant was transferred to a separate tube. Next, 750 μL ice-cold isopropanol was added to the supernatant, and the mixture was frozen at −80 °C for 15 min. The DNA was pelleted at top speed for 10 min at 4 °C, washed with 750 μL ice-cold 70% ethanol, and dissolved in 50 μL Tris-EDTA (TE) pH 8.0.

### 2.4. Lithium Acetate Transformations

The lithium acetate transformations were conducted as described previously, with some modifications [14]. Briefly, a 50 mL cell culture was grown to an OD_600_ of 1.0 and pelleted at 6000× *g* for 5 min. The pellet was washed 3 times with and resuspended in 1 mL sterile water. Next, 100 μL of cells were transferred to a sterile tube and pelleted to remove the supernatant. The pellet was resuspended in 360 μL of transformation mix (33% PEG 3350 with 0.1 M LiOAc, 100 μg salmon sperm carrier DNA, and 1 μg DNA), and vortexed briefly. The mixture was incubated with shaking at 39 °C and 300 rpm for 30 min, and then pelleted to remove the transformation mix. The pellet was resuspended in 1 mL of YPAD in a 1.5 mL Eppendorf tube, recovered for 2 h at 30 °C and 225 rpm, plated on the appropriate selective media, and incubated at 30 °C for 2 days.

### 2.5. Electroporation Conditions

Electroporation experiments were conducted as previously described [15], with slight modifications. Briefly, cells were grown to an OD_600_ of 1.5 in 50 mL YPAD and pelleted at 5000× *g* for 5 min. The pellet was resuspended in 10 mL 0.1 M LiOAc in 1×TE, pH 7.5, and incubated with shaking at 150 rpm and 30 °C for 1 h. Next, 250 μL 1 M DTT was added, and cells were returned to the incubator for 30 min. After adding 40 mL ice-cold sterile water, cells were pelleted, washed with 25 mL ice-cold sterile water, washed again with 5 mL ice-cold 1 M sorbitol, and resuspended in 250 μL 1 M sorbitol. Electro-competent cells were placed at −80 °C in 100 μL aliquots. When needed, cells were quickly thawed in a 37°C water bath and immediately placed on ice. Electroporation reactions were prepared in 0.2 cm cuvettes with 40 μL electro-competent cells, 5 μL DNA (1 μg), and 20 μg single-stranded carrier DNA, and conditions were 1.8 kV, 200 Ω, and 25 μF. Cells were collected from each cuvette in 1 mL of YPAD and allowed to recover in Eppendorf tubes for 2 h at 30 °C and 225 rpm. For each transformation, 20 μL were plated on the appropriate selective media and incubated at 30 °C for 2 days.

### 2.6. Locating the Insertion Site

*Metschnikowia borealis* cells were transformed with PCR-linearized *CaNAT1* flanked by 60 base-pair sequences of homology to the *HIS3* promoter and terminator, and DNA was isolated from 5 colonies by Alkaline Lysis as described above. DNA samples were digested with CfoI (Promega, Madison, WI, USA), with expected cut sites every ~600 base-pairs in the *M. borealis* genome and no cut sites within *CaNAT1*. Digested samples were then ligated using T4 DNA Ligase (New England BioLabsWhitby, ON, Canada). Qiagen Multiplex (Cat No. 206143) was used to PCR-amplify the unknown sequences surrounding the insertion site, using primers that bind within *CaNAT1* and that were oriented to amplify the surrounding sequence. PCR conditions were 95 °C melting for 30 s, 60 °C annealing for 90 s, and 72 °C extension for 90 s, cycled 35 times. PCR products were sent for sequencing at the DNA Sequencing Facility and Robarts Research Institute (London, ON, Canada). To identify insertion sites, sequences adjacent to the inserted marker were analyzed using BLASTn with Geneious 11.1.5 to search the *M. borealis* genome [16]. For secondary verification, primers were designed to anneal ~150 base-pairs upstream and downstream of the insertion site, and PCR was used to observe the insertion of the 693-base-pair cassette.

### 2.7. Mating Experiments

*M. borealis MATa* was transformed with *MbShBle*, and used in mating experiments with a *MATα* strain that had been transformed with *CaNAT1*. *MbShBle*-transformed *MATa* was thinly spread on 2% agar plates containing ½ strength glucose medium lacking histidine and uracil (Teknova, catalogue number C7221). *CaNAT1*-transformed *MATα* was then spread perpendicular to *MATa,* creating a grid with many sections where both strains were mixed. The cells were incubated for 2 days at room temperature. Approximately 20% of the cells were subsequently collected and resuspended in 25 mL of YPAD, and then grown at 30 °C for 6 h with shaking at 225 rpm. Next, 1 mL aliquots of culture were plated on YPAD containing 200 mg L^−1^ zeocin and 75 mg L^−1^ nourseothricin and incubated at 30 °C. After two days, resulting colonies were collected, serially diluted, and plated on YPAD with 200 mg L^−1^ zeocin and 75 mg L^−1^ nourseothricin. Purified recombinant colonies were grown overnight in YPAD, serially diluted, and plated alongside wildtype *MATa*, wildtype *MATα*, *MbShBle*-transformed *MATa*, and *CaNAT1*-transformed *MATα*.

## 3. Results

### 3.1. Identification of Antibiotics that Inhibit Growth of M. borealis

Before designing genetic cassettes (containing selection marker flanked by homology/promoter/terminator sequences) for transformation, it was necessary to identify antibiotics that inhibit the growth of *M. borealis* to enable us to choose appropriate antibiotic resistance markers. Cultures of *M. borealis MATa* and *MATα* were grown to OD_600_ of 1.5, pelleted, and resuspended to an OD_600_ of 3.0. Cultures were spot-diluted in 5 µL aliquots on YPAD agar plates with various concentrations of geneticin, zeocin, or nourseothricin (Appendix A); and it was determined that each antibiotic was effective at inhibiting growth at 400, 125, and 50 mg L^−1^, respectively. To further prevent the occurrence of spontaneous colonies, higher concentrations of nourseothricin (75 mg L^−1^ or 100 mg L^−1^) and zeocin (200 mg L^−1^) were used for selection.

### 3.2. Growth Rate of M. borealis

Many conventional yeast transformation methods require that a culture to be grown to a specific optical density at 600 nm; accordingly, it was necessary to determine the growth rate of *M. borealis*, to enable us to accurately calculate the growth time to the required optical density. Three cultures of each mating type of *M. borealis* (MATa and MATα) were grown to mid-log phase (OD_600_ = 1.0), and diluted in 50 mL of YPAD to an optical density of 0.065. Each culture was grown at 30 °C with shaking at 225 rpm, and OD_600_ was recorded at 1 h time points for 8 h. The average OD_600_ of the three cultures for each mating type was recorded, and the doubling time was calculated as 76 min for mating type a and 74 min for mating type α (Appendix A).

### 3.3. M. borealis Codon Usage

A series of transformation attempts were made with *M. borealis* using genetic cassettes containing the *NAT*, *KanMX*, and Sh *ble* genes; however, no colonies were ever obtained for any of the transformation attempts. While investigating factors that would prevent us from obtaining colonies of *M. borealis*, the question of genetic code became significant. Some species (including *M. bicuspidata* and *M. fructicola*) have been reported to recognize a CUG codon as a serine (yeast alternative nuclear code) [17,18], which fits within the current view that the Metschnikowiaceae belong to the CUG-Ser 1 clade of budding yeasts [19]. However, the only published literature regarding the genetic code of *M. pulcherrima* indicated that the species translates the CUG codon to leucine (standard code) [20]. The *NAT*, *KanMX*, and Sh *ble* cassettes used in the transformation attempts contain 8, 4, and 5 CUG codons in their open reading frames, respectively.

In an attempt to determine the genetic code of *M. borealis*, the coding sequence of *URA3* from *M. borealis* was translated with the ExPASy translate tool, using both the standard code and yeast alternative nuclear code, and aligned with the 30 top hits from a *URA3*-BLASTx search using MUSCLE (Figure 1A). A leucine aligned at a highly conserved serine or threonine position when *URA3* was translated with the standard genetic code; however, a serine aligned at that position when translated with the yeast alternative nuclear code. 

An examination of available Metschnikowiaceae genome sequences by tRNAscan-SE v2.0 [21] revealed that all species encoded at least one tRNA with a CAG anticodon. The retrieved tRNA sequences were analyzed through the tRNAscan-SE on-line web server, aimed at estimating the most probable translated amino acid. All *Metschnikowia* species that were examined had serine as the most probable translation with scores of 85 or more. Genome sequences for *Candida* aff. *bentonensis* or relatives are not available. Further evidence arose from aligning the sequence for tRNA_CAG_ from *M. borealis* with the tRNA_CAG_^Ser^ of related CUG-Ser clade yeast species (Figure 1B). The tRNA_CAG_ from *M. borealis* and other Metschnikowiaceae contain the serine discriminator base and identity elements, indicating that the tRNAs identified are in fact tRNA_CAG_^Ser^. No tRNA_CAG_^Leu^ could be identified in any of the *Metschnikowia* species that were analyzed, including *M. pulcherrima*.

### 3.4. Transformation of M. borealis with Codon-Optimized Cassettes

Optimizer was used to codon-optimize the *NAT*, *KanMX*, and Sh *ble* genes for expression in *M. borealis* based on the codon frequencies in three highly expressed yeast genes: *ADH1*, *TDH3*, and *ENO1*. The optimized genes were translated with the ExPASy translate tool, using both the standard code and yeast alternative nuclear code, and aligned with BLAST to ensure that the same protein product would be produced. No differences were found between the wildtype gene products and the optimized gene products, regardless of whether the optimized genes were translated with the standard code or yeast alternative nuclear code. The optimized *NAT* gene was substantially similar to *CaNAT1*, a gene optimized for expression in *Candida albicans* [22]. Due to its previous success in transforming *C. albicans*, another CUG-Ser species, *CaNAT1* was used in this experiment. Codon-optimized KanMX and Sh *ble* genes were called *MbKanMX* and *MbShble*.

Optimizer was used to codon-optimize the *NAT*, *KanMX*, and Sh *ble* genes for expression in *M. borealis* based on the codon frequencies in three highly expressed yeast genes: *ADH1*, *TDH3*, and *ENO1*. The optimized genes were translated with the ExPASy translate tool, using both the standard code and yeast alternative nuclear code, and aligned with BLAST to ensure that the same protein product would be produced. No differences were found between the wildtype gene products and the optimized gene products, regardless of whether the optimized genes were translated with the standard code or yeast alternative nuclear code. The optimized *NAT* gene was substantially similar to *CaNAT1*, a gene optimized for expression in *Candida albicans* [22]. Due to its previous success in transforming *C. albicans*, another CUG-Ser species, *CaNAT1* was used in this experiment. Codon-optimized KanMX and Sh *ble* genes were called *MbKanMX* and *MbShble*.

The second set of antibiotic resistance genes were created by reintroducing the same 8, 4, and 5 CUG codons found in the wildtype *NAT*, *KanMX*, and Sh *ble* genes, respectively, as a control for codon usage. All six codon-optimized genes (three in standard code and three in yeast alternative nuclear code) were ordered as separate G-block fragments from Integrated DNA Technologies. All genes were amplified with primers containing homologous hooks to the *M. borealis ADH1* promoter and terminator and were used to transform the species by electroporation. Transformants were plated on YPAD with appropriate concentrations of each antibiotic (75 mg L^−1^ nourseothricin for *CaNAT1*, 200 mg L^−1^ zeocin for *MbShBle*, and 400 mg L^−1^ geneticin for *MbKanMX*) alongside negative controls. Several colonies appeared for all samples that were transformed using the yeast alternative nuclear code, but no colonies appeared on any of the other plates (Figure 2). Similar results were seen using our best lithium acetate protocol. Selected transformants from the electroporation experiment were verified by genotyping (Appendix A). Transformation efficiencies were determined for each method by transforming *MATa* and *MATα* by each method in triplicate, and were calculated to be approximately 1600 and 817 CFUs per µg of DNA for electroporation, and 9 and 4 CFUs per µg of DNA for lithium acetate transformation, respectively (Appendix A).

The relatively high efficiency of electroporation was believed to be attributable to homologous recombination, because each cassette contained by 60 base-pairs of homology to the *ADH1* promoter and terminator. Based on this assumption, we attempted to generate histidine auxotrophs by transforming *M. borealis* with *CaNAT1* flanked by 60 base-pair sequences of the *HIS3* promoter and terminator. After transformation, ten colonies of each making type were spot-diluted and re-isolated, but all grew on media lacking histidine. Subsequent genotyping by PCR identified the presence of both *HIS3* and *CaNAT1*, but failed to obtain bands across the *HIS3* promoter and terminator junctions with *CaNAT1*: Bands that would be expected in the case of a successfully targeted knockout (Appendix A). Despite the random integration of our cassettes, double marker transformation experiments revealed that two cassettes were inserted in approximately 2% of transformants (Appendix A).

### 3.5. Location of Insertion Site

Subsequent screening of more than 30 *MATa* and 30 *MATα* transformants for *HIS3* knockouts failed to identify any successful insertions in the *HIS3* gene. To identify the correct insertion sites, DNA was isolated from 6 *MATα* transformants, digested with CfoI, circularized with T4 ligase, and the surrounding genomic DNA was PCR-amplified with primers binding within the inserted *CaNAT1* cassette (oriented to amplify the surrounding sequence) (Appendix A). We were successful in obtaining a PCR-fragment from the third *MATα* transformant (clone 3), and the amplified gDNA was sequenced. Analysis with BLASTn indicated that the *CaNAT1* cassette was inserted in scaffold bor-_s886 of *M. borealis* UWOPS 96-101.1, as annotated in GenBank (PRJNA312754), just upstream of the gene for peroxin-14. PCR was used to amplify this site in wildtype *M. borealis*, as well as all six isolated transformants, and the insertion was only observed at this site in clone 3.

### 3.6. Mating Experiments

To confirm that transformed-strains of *M. borealis MATa* and *MATα* were capable of successfully mating, *MbShBle*-transformed *MATa* and *CaNAT1*-transformed *MATα* were mated to develop ascospores (Figure 3A). Recombinant cells that contain both markers were isolated by serial dilution and genotyped to confirm that they contain both *MbShBle* and *CaNAT1* (Figure 3B). Mated cells were then plated alongside wildtype *MATa* and *MATα, MbShBle*-transformed *MATa*, and *CaNAT1*-transformed *MATα* on YPAD, YPAD with zeocin, YPAD with nourseothricin, and YPAD with nourseothricin and zeocin for comparison (Figure 3C).

### 3.7. Transformation of other CUG-Ser Yeasts

To determine if our electroporation method would yield transformants of other yeasts by random integration, we transformed strains of 21 yeast species with *CaNAT1* cassettes (in standard and yeast alternative nuclear code) flanked by the same 60-base-pair minimal *ADH1* promoter and terminator from *M. borealis* (Figure 4; Appendix A). Transformants were plated on YPAD with 75–200 mg L^−1^ nourseothricin and incubated at 27–30 °C for 24–72 h until colonies appeared. All strains yielded colonies for the alternative-coded *CaNAT1*, but only *S. cerevisiae* and *Candida* aff. *bentonensis* yielded colonies when transformed with standard-coded *CaNAT1* (Figure 4; Appendix A). It’s important to note that the alternative-coded *CaNAT1* lacks CUG codons and is therefore degenerate between the standard code and yeast alternative code; however, the standard-coded *CaNAT1* uses CUG codons to translate leucine and is only translatable to CUG-Leu yeasts.

## 4. Discussion

Through pioneering experiments, we were successful in determining the necessary conditions to transform *Metschnikowia borealis*, and the method was effective in many other yeast species. We were able to identify the tRNA_CAG_^Ser^ in *M. borealis*, as well as many other *Metschnikowia* species, and transformation experiments confirmed that many other—if not all—*Metschnikowia* species are also CUG-Ser 1 clade yeasts (Figure 2B and Figure 4). It is unclear why *M. pulcherrima* was previously reported to be a CUG-Leu yeast [9].

Once we determined that *M. borealis* utilizes the yeast alternative nuclear code, the species was successfully transformed with selectable markers that contain homologous hooks to the *HIS3* promoter and terminator, with the high efficiency. Transformants were first believed to have been obtained through a targeted knockout of *HIS3* by homologous recombination; however, all colonies grew on medium lacking histidine, and genotyping confirmed the presence of both the selectable marker and the wildtype *HIS3* gene. Additionally, we failed to observe any of the expected genotyping bands associated with the knockout junctions. This indicated that the cassettes were not incorporated into the genome by targeted integration, but by another mechanism with high efficiency. 

Although we only identified and confirm one insertion site (in clone 3), the *HIS3*-targeted cassette was found to insert in the site just upstream of the peroxin-14 gene in clone 3, and not in any of the other transformants tested (Appendix A). Therefore, the cassette must have inserted in different locations in the other transformants, indicating that *M. borealis* appears to integrate DNA cassettes randomly into its genome. Even though the species is not efficient at targeted integration, its high efficiency at random integration makes it simple to transform and incorporate exogenous sequences into the genome. 

Finally, our electroporation transformation method was successful across all CUG-Ser yeasts that were attempted, as well as *S. cerevisiae* and *Candida* aff. *bentonensis*. This confirms the suspicion [5] that *C. bentonensis* is not a member of the Metschnikowiaceae. The results of this study indicate that the species is in fact a CUG-Leu yeast. Fortunately, the *CaNAT1*, *MbShBle*, and *MbKanMX* genes used in our experiments lack the use of a CUG codon and are degenerate to the standard and yeast alternative nuclear code. Therefore, the markers would likely be universal to all CUG-Leu, CUG-Ser, and CUG-Ala yeasts [19]. 

Through the use of random integration, the methods outlined in this study make it possible to transform a large range of yeasts and would likely be successful in numerous additional species in a wide array of clades. Random integration can be used to introduce elements to many understudied yeast species to enable their study in more detail. In particular, it is now possible to design codon-optimized fluorescent nuclear-fusion proteins that can be used to elucidate the unusual meiotic process in *Metschnikowia* species that leads to the production of only two ascospores from a diploid zygote.

## Figures and Tables

**Figure 1 genes-10-00078-f001:**
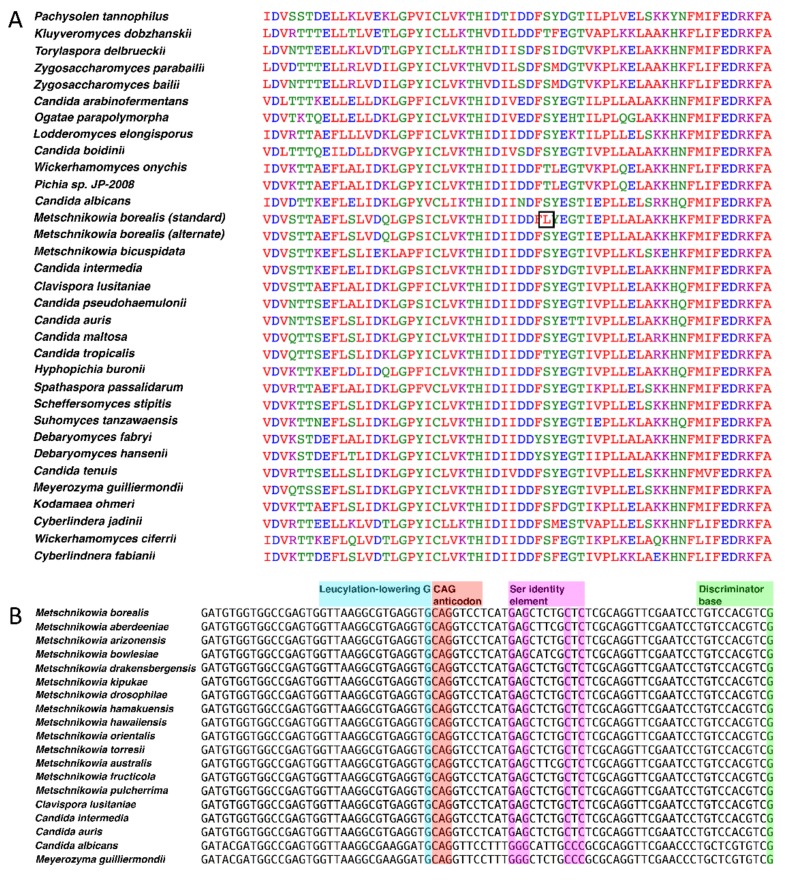
Bioinformatic analyses of the genetic code of *M. borealis*. (**A**) Sequence alignment of *URA3* (encodes Orotidine 5'-phosphate decarboxylase). The coding sequence of *URA3* from *M. borealis* was submitted as a query sequence to BLASTx, using the standard code for the predicted protein sequence. The protein sequences of the top 30 hits were aligned in MUSCLE with the *M. borealis URA3* gene, translated in both the standard code and yeast alternative code using the ExPASy translate tool. A CUG codon aligned at a highly conserved serine or threonine amino acid position, producing a leucine when *URA3* was translated using the standard code (black square). (**B**) tRNA_CAG_^Ser^ sequences aligned for many species of *Metschnikowia* and various CUG-Ser yeasts. Sequences were aligned in MEGA6 and annotated as described previously [18].

**Figure 2 genes-10-00078-f002:**
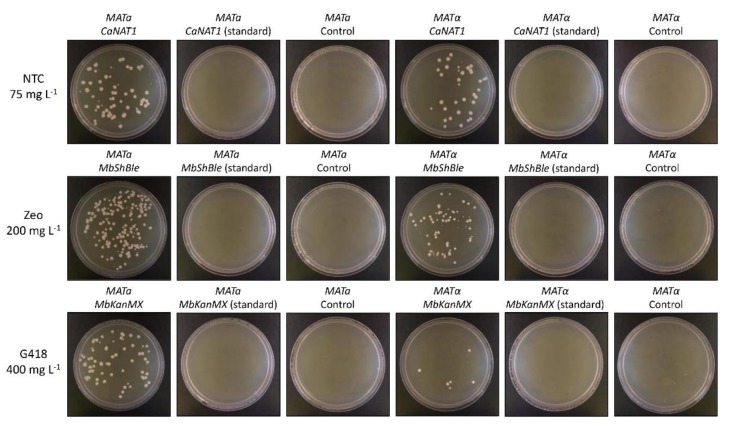
Electroporation of *M. borealis* with *CaNAT1*, *MbKanMX*, and *MbShBle* in standard and yeast alternative nuclear code. Electro-competent cells of *M. borealis* (*MATa* and *MATα*) were transformed with 1 µg of PCR-amplified *CaNAT1*, *MbShBle*, or *MbKanMx*, in standard or alternative code, flanked by 60 base-pair sequences of the *M. borealis ADH1* promoter and terminator, using the electroporation protocol outlined in this study. After recovery, 20 µg of each transformation was plated on YPAD with the appropriate antibiotic and incubated at 30 °C for two days. NTC = nourseothricin; Zeo = zeocin; G418 = geneticin.

**Figure 3 genes-10-00078-f003:**
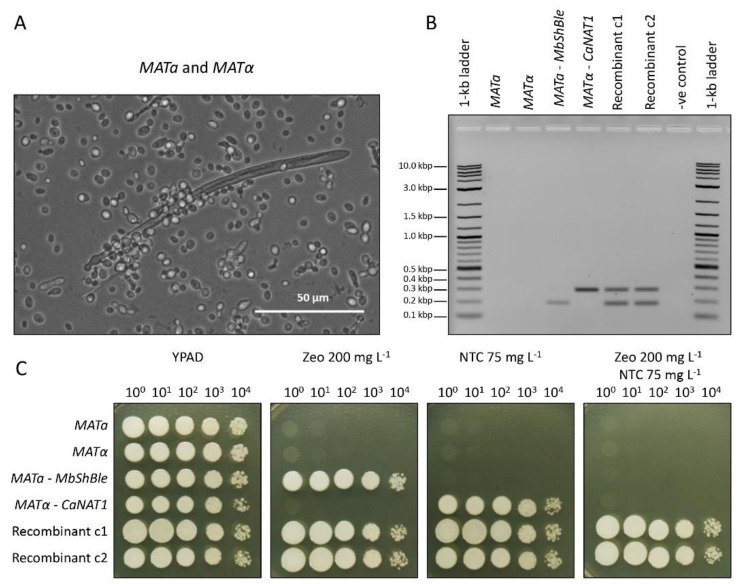
Mating experiments with marker-transformed *M. borealis*. (**A**) Photograph of a large mating ascospore. *MbShBle*-transformed *MATa* and *CaNAT1*-transformed *MATα* were mated for two days on ½-strength glucose broth lacking histidine and uracil (Teknova), collected in YPAD, and photographed under a light microscope. (**B**) Genotyping results of marker-transformed diploids. Recombinant colonies harbouring both selectable markers were purified by spot dilution on YPAD with 200 mg L^−1^ zeocin and 75 mg L^−1^ nourseothricin. Recombinant colonies were genotyped by Qiagen multiplex with primers that amplify the *CaNAT1* and *MbShBle* genes, alongside marker-transformed haploid strains and untransformed controls. *MbShBle* product size (**C**) Spot-dilutions of mated-colonies alongside transformed haploid strains and untransformed controls on YPAD, YPAD with 200 mg L^−1^ zeocin, YPAD with 75 mg L^-1^ nourseothricin, and YPAD with 200 mg L^−1^ zeocin and 75 mg L^−1^ nourseothricin. Plates were incubated at 30 °C for two days. NTC = nourseothricin; Zeo = zeocin; G418 = geneticin.

**Figure 4 genes-10-00078-f004:**
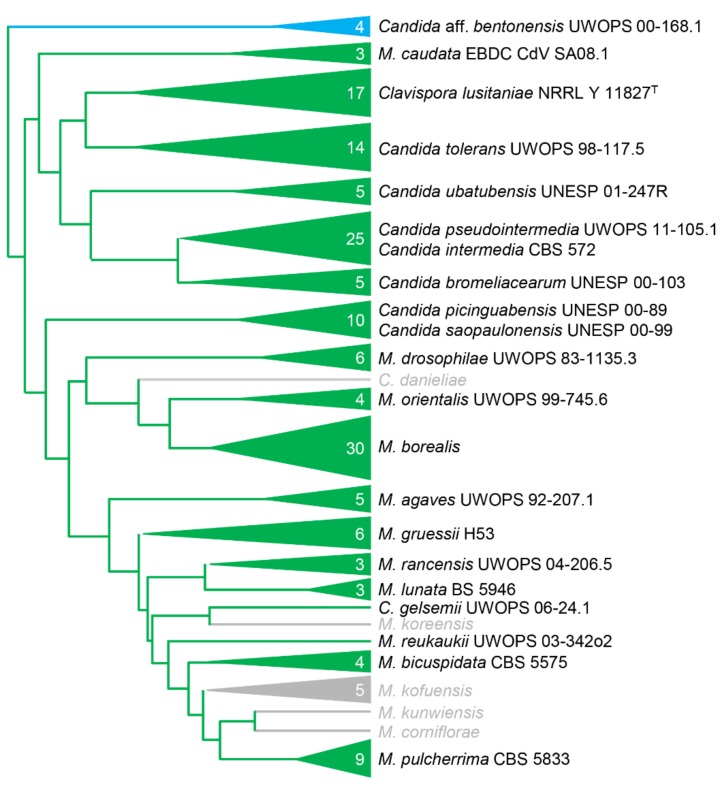
A phylogeny of the Metschnikowiaceae, modified from Lachance (2016) [5], showing the taxonomic range of successful transformations in the family. Representative strains were successfully transformed with only the alternative-coded *CaNAT1* construct (green) or with both the alternative-coded and the standard-coded *CaNAT1* (blue) construct. Species in grey were not examined.

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
