# Peer review of "Development of a Transformation Method for Metschnikowia borealis and other CUG-Serine Yeasts"

_genes, 2019, doi:10.3390/genes10020078_

Round 1
Reviewer 1 Report
In the reported Article manuscript, Development of a transformation method for Metschnikowia borealis and other CUG-serine yeasts, by Professor Bogumil J. Karas Lab, describe electroporation and lithium acetate methods using codon-optimized (CUG-serine) selectable markers in transforming Metschnikowia borealis, a large-spored Metschnikowia species. In preparing the genetic construct of M. borealis to harbor CaNAT and MbShBle genes, electroporation method generated transformants of the yeast by random integration, including other 19 yeast species. One yeast strain, M. borealis, with HIS3 cassette, was identified to inset in the site upstream of the peroxin-14 gene, an unexpected insertion site. A phylogeny of Metschenikowiaceae was verified by this approach and excluded Candida bentonensis from this taxonomic range. These engineered Metschenikowiaceae yeast strains will be quite useful in synthetic biology application and studying biology of Metschenikowiaceae.
This manuscript represents a great methodology and novel design of using CUG-serine codon optimized selection approach. The paper has general interests for synthetic biology community and reaches required novelty for publishing in Genes. Some minor issues need to be addressed still before publishing this work. (1) In Figure 1, URA3 gene was submitted for sequence alignment. What is the rationale for choosing this gene? How important is the conserved Ser/Thr’s role of URA3 gene product? (2) Clear explanation of random gene cassette insertion in the genome is not in the manuscript either in the literature. Can the CRISPER/CAS9 gene editing method solve this unexpected gene insertion?
Author Response
We thank the reviewer for these positive comments. Response to reviewer question is attached.

Reviewer 2 Report
A really nicely written paper!
a few pointers;
Inverse PCR is a great way to identify the integration sites. But it might be nice to do a southern or qPCR to get an idea of how many inserts your transformants have. With random integration it is not uncommon to get many copies of an insert, and I think it's important to define that. Multiple inserts can be as useful as single inserts depending on the application, it's certainly a useful parameter to know.
figure 3C looks great, it would be really nice if possible to also show segregation of the spores. Is it possible to dissect the spores and show phenotypic/genotypic segregation? maybe just a few plates worth to show that theoretically the cells are sexually reproducing and everything is functioning nicely - it would add weight to your rationale for doing this in the first place. While it's unlikely transformation would interrupt the ability to sporulate, if the goal is to truly understand meiosis functional sporulation would be pretty important.
for the text please pay attention to plurals, line 152 should read 2% agar plates containing 1/2 glucose medium lacking histidine and uracil (Teknova). - not media (which is plural).
Furthermore for this particular example the actual medium composition hasn't been written elsewhere in the manuscript so it should be included.
You're getting impressive transformation using electroporation which is great, but it's unclear to me from the text whether you tried to transform the codon optimized cassettes using Lithium acetate/PEG. If you didn't then it's unclear to me why you included a different initial transformation protocol to your final transformation protocol. If the cells are resistant to the lithium peg transformation protocol, it may be worth trying with different PEG chain lengths as different organisms require different PEG formulae. If they are not it might be worth comparing the two transformation protocols to see whether they differ in terms of homologous integration/transformation efficiency.
Author Response

(The authors gave the same response as above.)
